# Cardiac Hypertrophy May Be a Risk Factor for the Development and Severity of Glaucoma

**DOI:** 10.3390/biomedicines10030677

**Published:** 2022-03-15

**Authors:** Yukihisa Suzuki, Motohiro Kiyosawa

**Affiliations:** 1Department of Ophthalmology, Japan Community Health Care Organization, Mishima General Hospital, Shizuoka 411-0801, Japan; 2Research Team for Neuroimaging, Tokyo Metropolitan Institute of Gerontology, Tokyo 173-0015, Japan; 3Jiyugaoka Kiyosawa Eye Clinic, Tokyo 152-0035, Japan; nra12337@nifty.com

**Keywords:** bradycardia, electrocardiogram, glaucoma, left ventricular hypertrophy, visual field

## Abstract

The purpose of this study was to examine the relationship between glaucoma and cardiac abnormalities. We evaluated 581 patients with open-angle glaucoma (285 men and 296 women) and 595 individuals without glaucoma (273 men and 322 women). All of the participants underwent visual field testing using a Humphrey Visual Field Analyzer (30-2 program), an electrocardiogram (ECG), and blood pressure measurement. We examined the ECG abnormalities and other factors (age, intraocular pressure (IOP) and systemic hypertension) involved in the development and severity of glaucoma. Logistic regression analyses revealed significant correlations of glaucoma with IOP (OR = 1.43; 95% CI: 1.36–1.51; *p* < 0.00001), atrial fibrillation (OR = 2.02; 95% CI: 1.01–4.04; *p* = 0.04), left ventricular hypertrophy (LVH) (OR = 2.21; 95% CI: 1.15–4.25; *p* = 0.02), and bradycardia (OR = 2.19; 95% CI: 1.25–4.70; *p* = 0.02). Regression analyses revealed significant correlations of the mean deviation of the visual field with age (*t* = –6.22; 95% CI: −0.15, −0.08; *p* < 0.00001), IOP (*t* = −6.47; 95% CI: −0.42, −0.23; *p* < 0.00001), and LVH (*t* = −2.15; 95% CI: −3.36, −0.29; *p* = 0.02). Atrial fibrillation, LVH and bradycardia may decrease the cerebral blood flow, and may also affect the ocular blood flow. Cardiac abnormalities may be associated with the development and severity of glaucoma.

## 1. Introduction

Primary open-angle glaucoma is a chronic disease associated with intraocular pressure (IOP) that causes retinal nerve fiber layer loss and visual field (VF) defects [1]. It is well established that the main risk factor for glaucoma is IOP; however, there are cases with advanced visual field defects despite sufficient treatments for the lowering of IOP [2,3]. As a factor other than IOP, the involvement of vascular risk factors such as blood pressure or ocular perfusion pressure has been proposed as a pathological cause of glaucoma [2]. The presence of cardiac diseases can affect cerebral blood flow and ocular perfusion. Ischemic cardiovascular diseases have been suggested to be a risk factor for glaucoma progression [4]. Chen et al. [5] observed a significantly higher cumulative incidence of ischemic heart disease in the glaucoma group compared to the healthy group. The relationship between other cardiac diseases and glaucoma has not been examined, save for a few reports suggesting an association between atrial fibrillation (AF) and glaucoma [6]. However, AF [7] and left ventricular hypertrophy (LVH) [8,9] have been reported to reduce cerebral blood flow, and premature contractions involve increased fluctuations in the blood flow velocity in the middle cerebral artery from which the ophthalmic artery branches [10]. It is well known that the prevalence of glaucoma increases with age [11], and arrhythmias such as ischemic cardiac diseases and AF also increase with aging [12,13].

There are various types of ophthalmic solution that can be used for the treatment of glaucoma, and they have different mechanisms of action. Prostaglandin-related drugs—such as latanoprost, bimatoprost and toravoprost—mainly decrease IOP by increasing the outflow facility through an IOP-independent uveoscleral pathway, and they are most commonly used in current glaucoma treatments [14]. Brinzolamide and dorzolamide hydrochloride are a highly specific carbonic anhydrase inhibitor which lowers IOP by reducing the rate of aqueous humour formation [15]. Bunazosin hydrochloride is a potent and selective alpha1-adrenoceptor antagonist that has been used clinically an ocular hypotensive drug [16]. Brimonidine tartrate—an alpha1-adrenoceptor agonist—is an ophthalmic solution that lowers IOP by aqueous suppression, and by increasing uveoscleral outflow [17]. Ripasudil hydrochloride hydrate is a Rho-associated coiled-coil-containing protein kinase (ROCK) inhibitor that lowers IOP by increasing conventional aqueous outflow [18]. However, some eye drops cannot be used in patients with certain systemic diseases, and it is necessary to confirm the systemic diseases of each patient. Beta-adrenoceptor blocking agents (beta-blockers)—such as timolol, carteolol, and levobunolol—have long been used clinically as treatments for glaucoma despite their severe systemic side effects. Beta-blockers are contraindicated in patients with sinus bradycardia, second- or third-degree atrio-ventricular block, and asthma [19,20].

High IOP, aging, family history, and systemic hypotension [21] are among the factors reported to promote the development and progression of glaucoma; however, the relationship between glaucoma and cardiac abnormalities has not been studied. We hypothesized that cardiac diseases (AF, LVH, premature contraction, and bradycardia) are risk factors for the development of glaucoma, and we performed multivariable logistic regression analyses in patients with glaucoma and controls. Moreover, we also hypothesized that there is significant correlation between cardiac diseases and the severity of glaucoma, and we performed multiple regression analysis in patients with glaucoma. Moreover, we thought that knowing the proportion of patients with glaucoma contraindicated for beta-blockers would be useful for the management of patients. We believe that this study will provide new results for glaucoma patients with cardiac disease, which will help in future glaucoma treatment.

## 2. Materials and Methods

### 2.1. Standard Protocol Approvals, Registrations, and Patient Consent

The study protocol was approved by the institutional ethics committee of Japan Community Health Care Organization, Mishima General Hospital. All of the procedures conformed to the tenets of the Declaration of Helsinki. All of the measurements in this study were performed in the Japan Community Health Care Organization at Mishima General Hospital. Informed consent was obtained from all of the subjects before participation in this study. Informed consent was obtained from all of the subjects involved in the study. Written informed consent was obtained from the patients regarding the publication of this paper.

### 2.2. Subjects

Our study included 581 patients (285 men and 296 women, aged 71.6 ± 10.2 years) with open-angle glaucoma (Table 1). IOP was measured using a Goldman tonometer (Haag Streit, AT 900). The IOP in patients using glaucoma ophthalmic solution was measured after the discontinuation and washout of the ophthalmic solution. The VFs were assessed using the 30-2 program of the Humphrey Field Analyzer (Carl Zeiss Meditec, Dublin, CA, USA). The inclusion criteria included a diagnosis of primary open-angle glaucoma in accordance with the criteria reported by Foster et al. [1]. The glaucomatous VF defects were assessed based on the results of a glaucoma hemifield test, and were graded as abnormal when two clustered points were outside of the normal limits (*p* < 5%), or when a cluster of three contiguous points was below the 5% level on the pattern deviation plot [22]. Only the VF defects that were reproduced on at least two occasions were considered. The exclusion criteria were (1) patients aged <20 years, and (2) the presence or previous history of intraocular and intracranial diseases other than glaucoma. A classification proposed by Hodapp, Parrish and Anderson [22] is based on the severity of glaucoma (early stage, moderate stage, and severe stage). In each case, the severity of glaucoma of both eyes was determined, and the average of both eyes (the average stage of glaucoma) was calculated.

The control group consisted of 595 subjects (273 men and 322 women, aged 71.5 ± 7.9 years), who volunteered to participate in this study (Table 1). All of the control participants underwent ECG and eye examinations. Subjects with normal eye examinations were included in this group. The IOP and mean deviation of VF used for the statistical analysis were the average of the right and left eyes of each subject.

We excluded subjects taking beta-blocking medications and diuretics from the study, as those medicines can affect IOP. We defined a washout period of 8 weeks [23] for prostaglandins and 4 weeks [24] for other ophthalmic solutions. Participation in this study was abandoned for cases in which visual deterioration was predicted due to discontinuation of ophthalmic solutions. We excluded patients taking beta-blocking medications and diuretics from the study, as those medicines can affect IOP.

In a preliminary study, the prevalence of all ECG abnormalities was estimated to be 40% in the glaucoma group and 30% in the control group. The sample size required to detect a significant difference at a power of 80% and a significance level of 5% was 376 in both the glaucoma and control groups. Similarly, the prevalence of AF, LVH, and premature contraction was estimated to be 7% in the glaucoma group and 3.5% in the control group. The sample size required to detect a significant difference at a power of 80% and a significance level of 5% was 559 in both the glaucoma and control groups.

### 2.3. Recording of the Electrocardiogram, Blood Pressure Measurement and Data Processing

Standard 10-s 12-lead ECGs were obtained from all of the patients with glaucoma, and the control subjects. ECG abnormalities and heart rate were recorded. In the patients taking beta-bloker ophthalmic solutions, the ECG was recorded after the discontinuation of the ophthalmic solutions. From the results of the ECGs with automatic analysis, we examined the frequencies of ischemic abnormalities (infarction, ST elevation), atrial fibrillation (AF), QT prolongation, ST-T abnormality, left ventricular hypertrophy (LVH), an atrioventricular (A-V) block, a left bundle branch block (LBBB), premature contraction, a right bundle branch block (RBBB) and bradycardia in the glaucoma and control group. In the present study, the definition of bradycardia was 50/min or less.

We classified the ECG abnormalities into three levels of severity (mild, moderate, and severe) according to the standard criteria [25]. Severe abnormalities included ischemic abnormalities (myocardial infarction and ST elevation), AF, QT prolongation, abnormal Q wave, Mobitz 2 type of A-V block, and a three-degree A-V block. Moderate abnormalities included ST-T abnormality, LVH, LBBB, and the Wenkebach type of A-V block. Mild abnormalities included premature contraction, RBBB, bradycardia, sinus tachycardia, a first-degree A-V block, junctional rhythm, and intraventricular conduction disturbance. We examined the difference between the patient and control groups in the prevalence of each severity level of ECG abnormalities using the chi-square test.

After resting for 2 min, the blood pressure of each case was measured in the sitting position in our hospital. We defined systemic hypertension as a systolic blood pressure of 140 mmHg or higher or a diastolic blood pressure of 90 mmHg or higher, and we defined systemic hypotension as a systolic blood pressure of 100 mmHg or lower or a diastolic blood pressure of 60 mmHg or lower. In cases with white-coat hypertension, the results of the measurements may not reflect blood pressure in daily life. We tested the difference in the frequency of systemic hypertension and hypotension between the patient and control groups using the chi-square test.

### 2.4. Analysis of the Risk Factors for the Development and Severity of Glaucoma

We performed multivariable logistic regression analyses on all of the subjects (patients and control subjects) in order to investigate the risk factors for the development of glaucoma. Multivariable logistic regression analyses were performed to determine the association between the parameters (age, IOP, ECG abnormalities [ischemic abnormalities, AF, QT prolongation, LVH, ST-T abnormality, LBBB, premature contraction, RBBB and bradycardia], systemic hypertension and hypotension) and a diagnosis of glaucoma. Odds ratios (OR) and 95% confidence intervals (CI) were estimated for each factor. Statistical significance for all of the analyses was defined as *p* < 0.05.

Moreover, we performed a multiple regression analysis in patients with glaucoma in order to investigate the risk of the severity of the glaucoma. Multiple regression analyses were performed in order to determine the association between the mean deviation of VF and the parameters (age, IOP, ECG abnormalities [ischemic abnormalities, AF, QT prolongation, LVH, ST-T abnormalities, LBBB, premature contraction, RBBB and bradycardia], systemic hypertension and hypotension). The *t*-statistic and 95% CI were estimated for each factor. Statistical significance for all of the analyses was defined as *p* < 0.05.

These regression models were constructed following the identification of potential confounding variables. All of the analyses were conducted using EZR [26] (Saitama Medical Center, Jichi Medical University, Saitama, Japan), which is a graphical user interface for R (The R Foundation for Statistical Computing, Vienna, Austria).

## 3. Results

### 3.1. Frequency of Each of the ECG Abnormalities, Systemic Hypertension and Hypotension

Some ECG abnormalities were observed in 240 of the 581 patients with glaucoma and 184 of the 595 controls (Table 1). Severe ECG abnormalities were observed in 77 patients with glaucoma and 49 controls. Moderate and slight ECG abnormalities were observed in 137 and 111 patients with glaucoma, and in 132 and 98 controls, respectively. The number of cases in the glaucoma group and the control group for each EGC abnormality were as follows: 34 and 23 cases, respectively, of ischemic abnormalities, 30 and 18 cases of AF, 10 and 10 cases of QT prolongation, 86 and 70 cases of ST-T abnormality, 42 and 21 cases of LVH, 9 and 3 cases of LBBB, 45 and 32 cases of premature contraction, 59 and 54 cases of RBBB, and 28 and 19 cases of sinus bradycardia. The frequencies of mild ECG abnormalities (*p* = 0.01) and systemic hypertension (*p* = 0.005) were higher in the glaucoma group, while systemic hypotension was more frequent in the control group.

### 3.2. Risk Factors for the Development of Glaucoma

There was no difference in the frequency of severe and moderate ECG abnormalities. Logistic regression analyses revealed a significant correlation between glaucoma and IOP (OR = 1.43; 95% CI: 1.36–1.51; *p* < 0.00001), AF (OR = 2.02; 95% CI: 1.01–4.04; *p* = 0.04), LVH (OR = 2.21; 95% CI: 1.15–4.25; *p* = 0.02) and bradycardia (OR = 2.19; 95% CI: 1.25–4.70; *p* = 0.02) (Table 2). No significant correlation was observed between the prevalence of glaucoma and the other parameters.

### 3.3. Risk Factors for the Severity of Glaucoma

The regression analyses revealed a significant correlation between the MD of VF and age (*t* = −6.22; 95% CI: −0.15, −0.08; *p* < 0.00001), IOP (*t* = −6.47; 95% CI: −0.42, −0.23; *p* < 0.00001), and LVH (*t* = −2.15; 95% CI: −3.36, −0.29; *p* = 0.02) (Table 3). The frequency of mild ECG abnormalities was higher in the glaucoma group than the healthy group, and there were no differences in the frequency of severe and moderate ECG abnormalities between the glaucoma group and the control group.

### 3.4. Change of IOP Due to the Discontinuation of Ophthalmological Solutions, and Cases Excluded from the Study

Ophthalmic solution treatment was temporarily discontinued in 69 patients; the IOP before discontinuation was 14.4 ± 2.5 on the right and 14.4 ± 2.6 on the left, and the intraocular pressure after discontinuation was 19.5 ± 2.5 on the right and 19.6 ± 2.5 on the left. Three patients did not participate in the study for this reason.

## 4. Discussion

The frequencies of mild ECG abnormalities (*p* = 0.01) and systemic hypertension (*p* = 0.005) were higher in the glaucoma group, while systemic hypotension (*p* = 0.008) was more frequent in the control group. Similarly, ischemic abnormalities (*p* = 0.003), AF (*p* = 0.03), LVH (*p* = 0.003), ST-T abnormality (*p* = 0.03) and premature contraction (*p* = 0.04) also were higher in the glaucoma group. IOP, AF, LVH, and bradycardia may be associated with the development of glaucoma, whereas age, IOP, and LVH may be associated with the severity of glaucoma. About 4.8% of the patients with glaucoma had bradycardia, as an adverse effect of beta-blokers.

IOP is considered the main risk factor for the development of glaucoma, and is the only parameter subject to treatment [27]. In the Los Angeles Latino Eye Study—a population-based, prospective cohort study with over 4-years of follow-up—higher IOP was reported to be a risk factor for the development of open-angle glaucoma (OR per mmHg, 1.18) [28]. The results of several randomised controlled clinical trials have consistently attributed a 10% higher risk for both the development and the progression of the disease to each higher single mmHg [29] (Miglior 2013). Many long-term, randomized trials have shown the efficacy of lowering IOP, either by a percentage of the baseline, or to a specified level [30]. Diniz-Filho et al. [31] studied the association between IOP and the rates of retinal nerve fiber layer loss in patients with glaucoma using spectral-domain optical coherence tomography, and they observed that a 1 mmHg increase in the average IOP at the follow-up was associated with an additional average loss of 0.20 µm/year in progressing eyes. However, there are cases of glaucoma in which the disease progresses even though the expected decrease in IOP is achieved. In particular, in normal-tension glaucoma, the involvement of vascular dysregulation in the progression of glaucoma has been proposed [32].

Vascular dysregulation is suggested to be a main factor in the vascular pathophysiology of glaucomatous optic neuropathy [32]. Vascular dysregulation is defined as the inability of a tissue to maintain a constant blood supply despite changes in perfusion pressure. The ophthalmic blood flow is supplied through the ophthalmic artery, which branches from the internal carotid artery. Several studies have implicated vascular risk factors in the pathogenesis of glaucoma. Among them, blood pressure and ocular perfusion pressure have gained increasing attention. Some studies indicate that systemic hypertension is a risk factor for glaucoma [33,34]; however, some studies indicate that low systemic blood pressure is a risk factor for the development and progression of glaucoma. A direct and clear relationship between the blood pressure level and glaucomatous damage has not been established [35]. Population-based epidemiologic studies found strong relationships between low ocular perfusion pressure and open-angle glaucoma prevalence, as well as glaucoma incidence [35]. Clinical studies report similar associations between low perfusion pressure and glaucoma progression. Low systemic blood pressure combined with an elevated IOP reduces the perfusion pressure at the optic nerve head, which affects the volume of flow in eyes with an impaired autoregulatory system. Reduced flow can lead to ischaemic damage to the axons and the atrophy of the retinal ganglion cells, and consequentially to visual dysfunction [36]. In addition, some arrhythmias may affect cerebral and ocular blood flow. Decreased optic nerve blood flow has been reported in patients with glaucoma [37,38], and it has been presumed that abnormal perfusion and the subsequent ischemia of the optic nerve head play a major role in the glaucomatous damage. Krzyżanowska-Berkowska et al. [3] examined ocular blood flow biomarkers in the ophthalmic artery, the central retinal artery, and the nasal and temporal short posterior ciliary arteries in patients with open-angle glaucoma using color Doppler imaging; they observed that the ocular blood flow biomarkers in patients showed reduced peak systolic velocity and mean flow velocity compared with healthy controls. It is hypothesized that not only ocular blood flow but also cerebral blood flow itself is associated with the development and progression of glaucoma. Harris et al. [39] observed that the mean and peak systolic blood flow velocities in the middle cerebral artery were lower in glaucoma patients than in healthy subjects. Moreover, diffuse cerebral ischemic changes have been detected through magnetic resonance imaging in patients with normal-tension glaucoma [40]. From these findings, it seems that, for some patients, glaucomatous damage may be the ocular manifestation of a more widespread vascular condition involving the brain, rather than an isolated process of the eye and its immediate vasculature.

AF is a condition in which atrial contraction is irregular and inadequate due to abnormal electrical signals generated from a location other than the sinus node. AF creates a thrombus and causes a cerebral infarction by blocking a cerebral blood vessel [41]. It has been reported that even in patients with AF who do not develop obvious cerebral infarction, the cerebral blood flow decreases due to the microembolism of the brain caused by the fluctuations with each heartbeat, or by a decrease in stroke volume [41]. Zaleska-Żmijewska et al. [6] examined 79 patients with AF and 38 controls; glaucoma was confirmed in 35 (44%) individuals in the AF group and 5 (13%) individuals in the control group despite the equal IOP in both groups. LVH occurs as a compensatory response to an increased load on the myocardium, such as hypertension [8]. Normal LVH occurs in individuals with the normal cardiac function due to causes such as sports and pregnancy, but pathological LVH occurs due to aging and hypertension [19]. Decreased cardiac function due to hypertrophy leads to decreased cerebral blood flow. Bradycardia is a sinus node rhythm characterized by a low heart rate (50/min or less). Sierra et al. [42] examined the regional cerebral blood flow of patients with LVH using single photon emission computed tomography, and they observed regional cerebral blood flow ratio was significantly reduced in the striatum region of patients with LVH compared with controls. When the heart rate decreases below a certain level, cardiac output decreases and fainting may occur due to the insufficient blood flow to the brain. Solti et al. [43] observed a low cerebral blood flow in elderly patients with bradycardia, but it increased when the heart rate was normalized upon pacing. It is speculated that the cerebral blood flow in patients with bradycardia is chronically reduced.

It is known that glaucoma progresses with aging, and that the proportion of severe cases among the elderly is high [44]. It has now been well established that most visual functions decline with age, both for the fovea and the periphery [45]. Wild et al. [46] observed a loss of 0.7 dB per decade in the mean deviation of VF, and Adams et al. [47] estimated that the loss resulting from aging is approximately linear, at a pace of 0.6 dB per decade up to the age of 70, and greater after that age, as evaluated using frequency doubling technology perimetry. Nivison [48] examined a protein (mitofusin 2) involved in mitochondrial biogenesis, maintenance, and transport in retinal ganglion cells in mice with glaucoma; they observed that degenerated mitofusin 2 accumulates with age in retinal ganglion cells during glaucoma progression. On the other hand, it is known that the frequency of ECG abnormalities is high in the elderly, and the involvement of systemic hypertension, arteriosclerosis, and myocardial degeneration has been documented. It was recently found that the age-related thickening of blood vessels via angiotensin II causes the blood vessels to constrict, promoting elevated blood pressure, which may lead to LVH [49].

The A-V block is a conduction disorder from the atrium to the ventricle in the stimulation conduction system of the heart, and is classified into three degrees of severity. Second-degree A-V block includes the Wenckebach type and the Mobitz II type. The first-degree and Wenckebach type A-V block usually do not require treatment. Mobitz II type and third-degree A-V block may require sudden cardiac arrest and cardiac pacemaker implantation [50]. Beta-blockers are widely used in the treatment of hypertension and tachyarrhythmias (such as atrial fibrillation and sinus tachycardia) because they have the effect of suppressing atrioventricular conduction and reducing the heart rate. When the heart rate decreases below 40, syncope may occur [51]. Therefore, beta-blockers increase the probability of cardiac arrest not only in patients with bradyarrhythmias (Mobitz II type and third-degree A-V block) but also in patients with bradycardia without arrhythmias. Beta-blockers are also known to affect systemicity when they are used as an ophthalmic solution. Korte et al. [52] studied the cardiopulmonary effect of 0.5% ophthlalmic timolol, and compared it with that of intravenous timolol in healthy subjects. The peak concentration (C_max_) of ophthalmic timolol in plasma (1.14 ng/mL) was lower than that of intravenous timolol (4.85 ng/mL). However, the heart rate and blood pressure decreased after the administration of both ophthalmic and intravenous timolol. The heart rate decreased by 1 to 12 beats per minute (bpm) from the base level. Ishii et al. [53] also observed significant reductions in the heart rate (mean 8 bpm) and blood pressure after the administration of ophthalmic timolol compared with a placebo.

From these observations, we can see that ophthalmic beta-blockers not only reduce intraocular pressure but also affect the heart rate and blood pressure [52,53]. Moreover, beta-blocker ophthalmic solutions should not be prescribed in patients with sinus bradycardia or atrioventricular block, or in patients with systemic hypotension or orthostatic hypotension.

We used only ECGs in this study to diagnose cardiac diseases, but there is limitation regarding the diagnosis of LVH. The sensitivity and specificity of ECG to LVH is moderate, and it may not be adequately diagnosed (Devereux 1990; Goldberger 2017). LVH can be diagnosed more accurately by using echocardiography or cardiac magnetic resonance imaging.

## 5. Conclusions

Cardiac abnormalities in addition to IOP may be associated with the development and severity of glaucoma. Bradycardia is present in about 5% of glaucoma patients, and thus the ECG or heart rate of the patient should be checked before the prescription of beta-bloker ophthalmic solutions. We would like to investigate the effect of cardiac disease on the progression of glaucoma by observing the long-term course of glaucoma patients with cardiac disease in the future.

## Figures and Tables

**Table 1 biomedicines-10-00677-t001:** Demographic data of the patients with glaucoma and the controls.

		Glaucoma	Controls	*p* Value
Male:Female		285:296	273:322	0.3
Average year		71.6 ± 10.2	71.5 ± 7.9	0.8
IOP (mmHg)	Right	15.7 ± 4.6	12.6 ± 2.5	<0.00001
	Left	15.9 ± 4.2	12.7 ± 2.6	<0.00001
MD (dB)	Right	−4.35 ± 5.6	+0.40 ± 1.9	<0.00001
	Left	−4.88 ± 5.8	+0.27 ± 1.6	<0.00001
Systemic hypertension		199 (34.3%)	175 (29.4%)	0.0005
Systemic hypotension		51 (8.8%)	70 (11.8%)	0.0008
All ECG abnormalities		240 * (41.3%)	184 * (30.9%)	0.02
Severe abnormalities		77 * (10.7%)	49 * (8.2%)	0.1
Ischemic abnormalities		34 (5.9%)	23 (3.9%)	0.003
Atrial fibrillation		30 (5.2%)	18 (3.0%)	0.03
QT prolongation		10 (1.7%)	10 (1.7%)	0.8
3 degree A-V block		1 (0.2%)	0 (0%)	0.3
Abnormal Q		2 (0.3%)	0 (0%)	0.2
Moderate abnormalities		137 * (24.2%)	111 * (18.7%)	0.2
LVH		42 (7.2%)	21 (3.5%)	0.002
ST-T abnormality		86 (14.8%)	70 (11.8%)	0.03
Left bundle branch block		9 (1.5%)	3 (0.5%)	0.06
Mild abnormalities		132 * (29.2%)	98 * (16.5%)	0.01
Premature contraction		45 (7.7%)	32 (5.4%)	0.04
Right bundle branch block		59 (10.2%)	54 (9.1%)	0.3
Bradycardia		28 (4.8%)	19 (3.2%)	0.08

* There were cases where the same level of abnormality was duplicated. IOP, intraocular pressure; MD, mean deviation; ECG, electrocardiogram; A-V, atrioventricular; LVH, left ventricular hypertrophy.

**Table 2 biomedicines-10-00677-t002:** Odds ratio of each factor related to glaucoma development.

Factor	Odds Ratio	95% CI	*p* Value
Age	1.01	0.99, 1.02	0.3
IOP	1.43	1.36, 1.51	<0.00001
Ischemic abnormalities	1.10	0.60, 2.04	0.8
Atrial fibrillation	2.02	1.01, 4.04	0.04
QT prolongation	1.27	0.45, 3.59	0.7
LVH	2.21	1.15, 4.25	0.02
ST-T abnormality	0.94	0.62, 1.43	0.8
Left bundle branch block	1.71	0.42, 7.01	0.5
Premature contraction	1.71	0.98, 2.98	0.06
Right bundle branch block	1.07	0.68, 1.68	0.8
Bradycardia	2.19	1.25, 4.70	0.02
Systemic hypertension	0.97	0.73, 1.29	0.8
Systemic hypotension	0.96	0.62, 1.49	0.9

IOP, intraocular pressure; LVH, left ventricular hypertrophy.

**Table 3 biomedicines-10-00677-t003:** Factors associated with glaucoma severity.

Factor	*t*-Statistic	95% CI	*p* Value
Age	−6.22	−0.15, −0.08	<0.00001
IOP	−6.47	−0.42, −0.23	<0.00001
Ischemic abnormalities	0.01	−1.62, 1.77	0.9
Atrial fibrillation	−1.40	−3.07, 0.56	0.2
QT prolongation	−1.03	−4.67, 1.45	0.3
LVH	−2.15	−3.36, −0.29	0.02
ST-T abnormality	−0.85	−1.71, 0.67	0.4
Left bundle branch block	−1.53	−5.64, 0.68	0.1
Premature contraction	−0.38	−1.77, 1.20	0.1
Right bundle branch block	0.29	−1.13, 1.52	0.8
Bradycardia	−0.20	−2.04, 1.63	0.8
Systemic hypertension	−0.45	−1.05, 0.66	0.7
Systemic hypotension	0.48	−0.93, 1.89	0.7

IOP, intraocular pressure; LVH, left ventricular hypertrophy.

## Data Availability

All of the data used for the analysis are presented in the tables in this article. The data will be shared after ethics approval if requested by other investigators for the purposes of replicating the results.

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
