# Peer review of "Cardiac Hypertrophy May Be a Risk Factor for the Development and Severity of Glaucoma"

_biomedicines, 2022, doi:10.3390/biomedicines10030677_

Round 1
Reviewer 1 Report
The authors presented an interesting study examining the relationship between cardiac abnormalities and glaucoma. The manuscript is with merit but it presents several limitations that should be addressed before publication can be considered.
- Abstract/entire manuscript
- Please delete the space before and after the signs “=”, “<” etc., in the entire manuscript (for example at line 17 “OR = 1.43” should be replaced by “OR=1.43”)
- In the abstract and main document, the authors should provide additional conclusive insights regarding the potential applications of their findings in the clinical practice of ophthalmologists/cardiologists and general practitioners and suggest the importance and direction of future studies investigating the relationship between glaucoma and cardiac diseases.
- In the abstract and main document, the terms “hypotension” and “hypertension” should be replaced by “systemic hypotension/hypertension”
- Introduction
- The introduction should be expanded in its different sections and the authors should provide additional evidence with corresponding citations of their statement, as examples:
- LINE 36 “The presence of cardiac diseases can affect cerebral blood flow and ocular perfusion”: the authors should expand this statement and provide additional information and findings and related citations.
- LINE 41- 42: the authors should expand this sections and list and brieflu described all the categories of IOP lowering drugs available; also, the authors should elaborate more regarding the side effects of beta blockers in relation to cardiovascular diseases.
- The last paragraph of the introduction should be expanded, and the authors should state clearly, after the statements regarding their hypothesis, the aim of their study (as last sentence of the introduction)
- The introduction should be expanded in its different sections and the authors should provide additional evidence with corresponding citations of their statement, as examples:
- Material and methods
- The information regarding the demografic of the patients and the reference to Table 1 and table 1 itself should be placed in the “results” section, not in the methods. In the methods only the description of the methods should be provided, not data on the results
- No indication is given in the manuscript body related the compliance of the study to the Declaration of Helsinki. The authors should indicate if they received IRB approval for the study, with details. Also, no information is given related to the informed consent of the participants. If possible per the journal guidelines, these information should be provided also in the manuscript and not only in the statements at the end.
- The legend of Table 1 states “Demographic data of patients with amblyopia” – the table and related results should be presented in the “results” sections; in addition, what the author mean by amblyopia? This is a specific eye condition that is not otherwise mentioned anywhere else in the body of the manuscript, please clarify.
- The authors state that the IOP in patients using glaucoma ophthalmic solution was measured after discontinuation and washout of the ophthalmic solution – the authors should provide specific details (timeline, duration) of such discontinuation. Also, how do the authors address the risk of IOP increase after drug discontinuation? Was the IOP measured before and after the discontinuation? Any of the study patients had to discontinue the study for IOP increase?
- Were the study participants evaluated for systemic use of drugs or medications that may have influenced the results? Were there specific criteria of inclusion/exclusion for use of systemic medications?
- The authors state that they evaluate the risk of “progression” of glaucoma: how did they evaluate glaucoma progression? Were there follow up visits that allowed the evaluation of progression (functional/structural) over a period of time? The authors should provide specific details.
- Statistics: The author should provide a statistical power estimation for their study or at least some justification of the study n and add it to the methods
- Results
- The first part of the results should describe the study population in terms of demographics and other characteristics – all these details should be moved form the “methods” section to the “results” sections.
- In the results section related to “the risk factors for progression” the authors mention the disease severity: in that case, the authors should not use the term “progression” (that can only be evaluated over time) but “severity” and revise correspondingly the tnire manuscript and table 3 legend
- Discussion
- The discussion should be re-written and re-organize in order to provide the findings of this study and discuss such results in comparison to what is known from the literature, as it currently stands the authors provide a lot of information but there is not a clear link with the results of their study:
- First paragraph: The authors should revise the wording and indicate in the text if the differences between groups were or not “statistically significant” and not only cite the dofferences in percentage between the two groups
- Second paragraph: the authors should provide more details about the vascular risk factors in glaucoma with corresponding citations, and the anatomic description of the blood supply to the eye should be described first when introducing the topic of the vascular risk factors.
- The discussion should be re-organized in order to both provide information about the different risk factors (IOP, vascular risk factors) for glaucoma and different cardiac conditions there should be a link to the finding of this specific study. The authors provide details of the different conditions, but there should be a corresponding discussion and reference to this study and their findings.
- Lines 310-305: it is not clear why this paragraph is located in this position and to which specific finding of the study is related to.
- Conclusion
- The authors should rephrase the conclusion considering the concept of severity of glaucoma instead of progression (per my comment above)
- The authors should elaborate more on the clinical applications of their study and future studies needed in this field.
- The discussion should be re-written and re-organize in order to provide the findings of this study and discuss such results in comparison to what is known from the literature, as it currently stands the authors provide a lot of information but there is not a clear link with the results of their study:
Author Response
Dear reviewer 1
Thank for your detailed review and kind comments.
biomedicines-1587966, MS TITLE: Cardiac hypertrophy may be a risk factor for the development and progression of glaucoma
Yukihisa Suzuki, and Motohiro Kiyosawa
Comments of Reviewer
Please delete the space before and after the signs “=”, “<” etc., in the entire manuscript (for example at line 17 “OR = 1.43” should be replaced by “OR=1.43”)
Response: We deleted the space before and after the sign in the entire manuscript.
In the abstract and main document, the authors should provide additional conclusive insights regarding the potential applications of their findings in the clinical practice of ophthalmologists/cardiologists and general practitioners and suggest the importance and direction of future studies investigating the relationship between glaucoma and cardiac diseases.
Response: We added descriptions about clinical applications in the future in the last of abstract and conclusion.
In the abstract and main document, the terms “hypotension” and “hypertension” should be replaced by “systemic hypotension/hypertension”
Response: We rewrote “hypotension” and “hypertension” into “systemic hypotension/hypertension” in the entire manuscript.
Introduction
The introduction should be expanded in its different sections and the authors should provide additional evidence with corresponding citations of their statement, as examples:
LINE 36 “The presence of cardiac diseases can affect cerebral blood flow and ocular perfusion”: the authors should expand this statement and provide additional information and findings and related citations.
Response: We added the descriptions of relation between ischemic heart diseases and glaucoma and relation between cardiac diseases and cerebral blood flow in first paragraph of Introduction section.
LINE 41- 42: the authors should expand this sections and list and brieflu described all the categories of IOP lowering drugs available; also, the authors should elaborate more regarding the side effects of beta blockers in relation to cardiovascular diseases.
Response: The types of glaucoma ophthalmic solutions used clinically include prostaglandin-related drugs, carbonic anhydrase inhibitor, alpha1-adrenoceptor antagonist, alpha1-adrenoceptor agonist and rho-associated coiled-coil-containing protein kinase (ROCK) inhibitor in addition to beta-blockers, and we added descriptions in second paragraph of Introduction section.
The last paragraph of the introduction should be expanded, and the authors should state clearly, after the statements regarding their hypothesis, the aim of their study (as last sentence of the introduction)
Response: We added the sentence “We believe that this study will provide new results for glaucoma patients with cardiac disease, which will help in the future glaucoma treatment.“ in the last of Introduction section.
Material and methods
The information regarding the demografic of the patients and the reference to Table 1 and table 1 itself should be placed in the “results” section, not in the methods. In the methods only the description of the methods should be provided, not data on the results
Response: We moved Table 1 and related descriptions to the Results section.
No indication is given in the manuscript body related the compliance of the study to the Declaration of Helsinki. The authors should indicate if they received IRB approval for the study, with details. Also, no information is given related to the informed consent of the participants. If possible per the journal guidelines, these information should be provided also in the manuscript and not only in the statements at the end.
Response: We added the descriptions about Standard protocol approvals, registrations, and patient consents in Materials and Methods section (2.4. Standard protocol approvals, registrations, and patient consents section).
The legend of Table 1 states “Demographic data of patients with amblyopia” – the table and related results should be presented in the “results” sections; in addition, what the author mean by amblyopia? This is a specific eye condition that is not otherwise mentioned anywhere else in the body of the manuscript, please clarify.
Response: The description of amblyopia is a simple mistake, correctly glaucoma and Controls. We rewrote the legend of Table 1 into “Demographic data of patients with glaucoma and Controls”.
The authors state that the IOP in patients using glaucoma ophthalmic solution was measured after discontinuation and washout of the ophthalmic solution – the authors should provide specific details (timeline, duration) of such discontinuation. Also, how do the authors address the risk of IOP increase after drug discontinuation? Was the IOP measured before and after the discontinuation? Any of the study patients had to discontinue the study for IOP increase?
Response: We defined a washout period of 8 weeks (Sharpe 2008) for prostaglandins and 4 weeks (Janulevicienë 2004) for other ophthalmic solutions. Participation in this study was abandoned for cases in which visual deterioration was predicted due to discontinuation of ophthalmic solusions. Ophthalmic solution treatment was temporarily discontinued in 69 patients, and the IOP before discontinuation was 14.4 ± 2.5 on the right and 14.4 ± 2.6 on the left, and the intraocular pressure after discontinuation was 19.5 ± 2.5 on the right and 19.6 ± 2.5 on the left. Three patients did not participate in the study for this reason. We added these descriptions in Material and methods section (2.1. Subjects section).
Were the study participants evaluated for systemic use of drugs or medications that may have influenced the results? Were there specific criteria of inclusion/exclusion for use of systemic medications?
Response: We excluded subjects taking beta-blocking medications and diuretics from the study, because those medicines can affect IOP. We added these descriptions in Material and methods section. (2.1. Subjects section).
The authors state that they evaluate the risk of “progression” of glaucoma: how did they evaluate glaucoma progression? Were there follow up visits that allowed the evaluation of progression (functional/structural) over a period of time? The authors should provide specific details.
Response: We have not been able to assess the risk of "progression" of glaucoma because we have not examined the long-term course of each patient. We examined the severity of glaucoma, not the risk of progression.
Statistics: The author should provide a statistical power estimation for their study or at least some justification of the study n and add it to the methods
Response: In a preliminary study, the prevalence of All ECG abnormalities was estimated to be 40% in the glaucoma group and 30% in the control group. The sample size required to detect a significant difference at a power of 80% and a significance level of 5% was 376 in both glaucoma and control groups. Similarly, the prevalence of AF, LVH, and premature contraction was estimated to be 7% in the glaucoma group and 3.5% in the control group. The sample size required to detect a significant difference at a power of 80% and a significance level of 5% was 559 in both glaucoma and control groups. We added these descriptions in Material and methods section. (2.1. Subjects section).
The first part of the results should describe the study population in terms of demographics and other characteristics – all these details should be moved form the “methods” section to the “results” sections.
Response: Demographic data of patients with glaucoma and controls was moved into Results section.
In the results section related to “the risk factors for progression” the authors mention the disease severity: in that case, the authors should not use the term “progression” (that can only be evaluated over time) but “severity” and revise correspondingly the tnire manuscript and table 3 legend
Response: We rewrote “progression” into “severity” in manuscript and table 3.
The discussion should be re-written and re-organize in order to provide the findings of this study and discuss such results in comparison to what is known from the literature, as it currently stands the authors provide a lot of information but there is not a clear link with the results of their study
Response: We reorganized the description of intraocular pressure in the second paragraph and the description of vascular risk factors in the third paragraph in Discussion section.
First paragraph: The authors should revise the wording and indicate in the text if the differences between groups were or not “statistically significant” and not only cite the dofferences in percentage between the two groups
Response: We added the difference in the frequency of each ECG abnormality (ischemic abnormalities [P=0.003], AF [P=0.03], LVH [P=0.003], ST-T abnormality [P=0.03] and premature contraction [P=0.04]) between groups with the P-value.
Second paragraph: the authors should provide more details about the vascular risk factors in glaucoma with corresponding citations, and the anatomic description of the blood supply to the eye should be described first when introducing the topic of the vascular risk factors.
Response: We added descriptions about vascular dysregulation, ophthalmic blood flow, and systemic blood pressure in third paragraph of Discussion section.
The discussion should be re-organized in order to both provide information about the different risk factors (IOP, vascular risk factors) for glaucoma and different cardiac conditions there should be a link to the finding of this specific study. The authors provide details of the different conditions, but there should be a corresponding discussion and reference to this study and their findings.
Response: We added descriptions about IOP, vascular risk factors and cardiac diseases in second, third and fourth paragraph of Discussion section.
Lines 310-305: it is not clear why this paragraph is located in this position and to which specific finding of the study is related to.
Response: We moved this description to Materials and methods section (2.2. Recording of electrocardiogram, blood pressure measurement and data processing section).
Conclusion
The authors should rephrase the conclusion considering the concept of severity of glaucoma instead of progression (per my comment above)
Response: We rewrote “progression” into “severity” in Conclusion section.
The authors should elaborate more on the clinical applications of their study and future studies needed in this field.
Response: We would like to investigate the effect of cardiac disease on the progression of glaucoma by observing the long-term course of glaucoma patients with cardiac disease in the future. We added the sentence in Conclusion section.
Reviewer 2 Report
This study was to examine the relation between glaucoma and cardiac abnormalities by measurement of electrocardiogram (ECG), blood pressure measurement, Visual Field Analyzer, and IOP. It is an interesting study and may be important of patients with glaucoma. However, there are some points should be clarified.
- The term “Left ventricular hypertrophy (LVH)” may be more adequate than “cardiac hypertrophy”.
- The limitations of the ECG relate to its moderate sensitivity or specificity depending upon which of the many proposed sets of diagnostic criteria are applied [1,2]. Therefore, because of these ECG limitations, LVH is most reliably identified on imaging with either echocardiography or cardiac magnetic resonance imaging.
- Goldberger AL, Goldberger ZD, Shvilkin A. Goldberger's Clinical Electrocardiography: A Simplified Approach, 9th ed, Elsevier/Saunders, Philadelphia 2017.
- Devereux RB. Is the electrocardiogram still useful for detection of left ventricular hypertrophy? Circulation 1990; 81:1144.
- In the abstract: “Atrial fibrillation, cardiac hypertrophy and bradycardia may decrease cerebral blood flow and may also affect ocular blood flow. Not only ocular blood flow but also cerebral blood flow may be associated with the pathology of glaucoma.” à
“Not only ocular blood flow but also cerebral blood flow may be associated with the pathology of glaucoma” can not be proved by this study.
Author Response
Dear reviewer 2
Thank for your detailed review and kind comments.
biomedicines-1587966, MS TITLE: Cardiac hypertrophy may be a risk factor for the development and progression of glaucoma
Yukihisa Suzuki, and Motohiro Kiyosawa
Comments of Reviewer
The term “Left ventricular hypertrophy (LVH)” may be more adequate than “cardiac hypertrophy”.
Response: We rewrote “cardiac hypertrophy” into “left ventricular hypertrophy (LVH)” in the manuscript.
The limitations of the ECG relate to its moderate sensitivity or specificity depending upon which of the many proposed sets of diagnostic criteria are applied [1,2]. Therefore, because of these ECG limitations, LVH is most reliably identified on imaging with either echocardiography or cardiac magnetic resonance imaging.
Goldberger AL, Goldberger ZD, Shvilkin A. Goldberger's Clinical Electrocardiography: A Simplified Approach, 9th ed, Elsevier/Saunders, Philadelphia 2017.
Devereux RB. Is the electrocardiogram still useful for detection of left ventricular hypertrophy? Circulation 1990; 81:1144.
Response: We added the descriptions about the limitation of the ECG in the last paragraph of Discussion section.
In the abstract: “Atrial fibrillation, cardiac hypertrophy and bradycardia may decrease cerebral blood flow and may also affect ocular blood flow. Not only ocular blood flow but also cerebral blood flow may be associated with the pathology of glaucoma.” “Not only ocular blood flow but also cerebral blood flow may be associated with the pathology of glaucoma” can not be proved by this study.
Response: We deleted the description of “Not only ocular blood flow but also cerebral blood flow may be associated with the pathology of glaucoma.” in the Abstract.
Round 2
Reviewer 1 Report
The authors addressed most of the comments. However, there are still some comments to be addressed:
- Introduction: The aim of the study should be clearly defined in the last paragraph of the introduction; the authors added a sentence in the revised manuscript that do not indicate the specific purpose (such as: "the aim of the study in to investigate the relationship between glaucoma and cardiac abnormalities"
- The authors provided information about the details on the washout period for the IOP lowering medications. The general information about the washout should be provided in the methods, while the specific details on the number of patients/IOP values should be moved to the “results” section.
- The paragraph 2.4 should be moved at the beginning of the Methods
Author Response
Dear reviewer 1
Thank for your detailed review and kind comments.
biomedicines-1587966, MS TITLE: Cardiac hypertrophy may be a risk factor for the development and severity of glaucoma
Yukihisa Suzuki, and Motohiro Kiyosawa
Comments of Reviewer
Introduction: The aim of the study should be clearly defined in the last paragraph of the introduction; the authors added a sentence in the revised manuscript that do not indicate the specific purpose (such as: "the aim of the study in to investigate the relationship between glaucoma and cardiac abnormalities"
Response: We rewrote the aim of this study into "We hypothesized that cardiac diseases (AF, LVH, premature contraction, and bradycardia) are risk factors for the development of glaucoma, and we performed multivariable logistic regression analyzes in patients with glaucoma and controls. Moreover, we also hypothesized that there are significant correlation between cardiac diseases and the severity of glaucoma, and we performed multiple regression analysis in patients with glaucoma."
The authors provided information about the details on the washout period for the IOP lowering medications. The general information about the washout should be provided in the methods, while the specific details on the number of patients/IOP values should be moved to the “results” section.
Response: We moved these descriptions into the last of Materials and Methods section.
The paragraph 2.4 should be moved at the beginning of the Methods
Response: We moved the paragraph of “Standard protocol approvals, registrations, and patient consents” into the beginning of Materials and Methods section.
Reviewer 2 Report
the manuscript has been sufficiently improved according to my suggestion
Author Response
Thank for your kind comments.
Round 3
Reviewer 1 Report
The revised manuscript is definitively been improved by the authors that addressed the majority of the comments but one.
The authors revised the last part of the introduction indicating their hypotheses but they did not included in the last paragraph a simple and explicit sentence indicating the specific purpose of the study – for the reader the aim of the study should be please clearly stated in addition to the study hypothesis: therefore please revise and add a sentence related to the purpose of the study, such as: "the aim of the study is to investigate the relationship between glaucoma and cardiac abnormalities" or similar.